# Lung Adenocarcinoma Exhibiting Thanatosomes (Hyaline Bodies), Cytoplasmic Clearing, and Nuclear Pleomorphism, with a *KRAS* Mutation

**DOI:** 10.3390/diagnostics15070894

**Published:** 2025-04-01

**Authors:** Mitsuhiro Tachibana, Yutaro Ito, Ryo Fujikawa, Kei Tsukamoto, Masahiro Uehara, Jun Kobayashi, Takuo Hayashi

**Affiliations:** 1Department of Diagnostic Pathology, Shimada General Medical Center, Shimada 427-8502, Japan; 2Department of Respiratory Medicine, Shimada General Medical Center, Shimada 427-8502, Japan; 3Department of Respiratory Surgery, Shimada General Medical Center, Shimada 427-8502, Japan; 4Department of Diagnostic Radiology, Shimada General Medical Center, Shimada 427-8502, Japan; 5Department of Human Pathology, Juntendo University Graduate School of Medicine, Bunkyo-ku, Tokyo 113-8431, Japan; tkhyz@juntendo.ac.jp

**Keywords:** *KRAS*, *MET* exon 14 skipping, thanatosomes, hyaline globules, lung adenocarcinoma, TTF-1

## Abstract

Since epidermal growth factor receptor (EGFR) tyrosine kinase inhibitors were introduced in 2004, various driver gene mutations have been identified in non-small cell lung cancer, particularly adenocarcinoma, where mutations are typically mutually exclusive. *EGFR* and *Kirsten rat sarcoma viral oncogene (KRAS)* mutations are most prevalent in Japan, with routine testing now standard. However, hematoxylin and eosin staining often fails to detect mutations, except in cases such as *ALK* fusion lung cancer. We report a 76-year-old non-smoking Japanese woman diagnosed with adenocarcinoma confirmed as *KRAS* G12D/S-positive. Histological features, including thanatosomes (hyaline globules), nuclear pleomorphism, and cytoplasmic clearing, may aid in identifying mutations. Numerous thanatosomes were identified, some containing nuclear dust. Thanatosomes revealed periodic acid–Schiff reactivity with diastase resistance, fuchsinophilia with Masson’s trichrome stain, and dark blue-black color with Mallory’s PTAH stain. This is the first report linking thanatosomes in *KRAS*-mutant pulmonary adenocarcinoma to apoptosis via cleaved caspase-3 staining.

**Figure 1 diagnostics-15-00894-f001:**
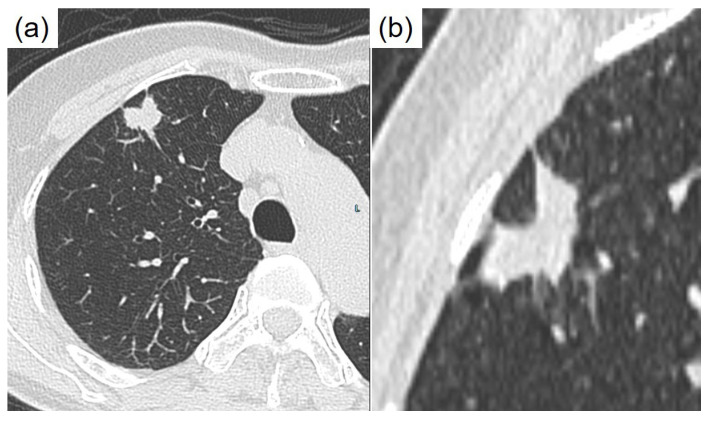
Clinical imaging findings. Enhanced computed tomographic images. Since the dramatic therapeutic effects of epidermal growth factor receptor (EGFR) tyrosine kinase inhibitors with activating alterations of the *EGFR* gene were first reported in 2004 [1], numerous driver gene variants have been identified as therapeutic targets in non-small cell lung cancer, particularly in adenocarcinoma [2,3,4]. In lung adenocarcinoma, driver gene variants occur mutually exclusively [5]. According to the National Cancer Center (NCC) Japan, most mutations in this country involve *EGFR* and *Kirsten rat sarcoma viral oncogene (KRAS)*, whereas other mutations are less common [5]. In clinical practice for lung cancer, the routine search for driver gene mutations has become standard with tests such as the Amoy Dx^®^ Lung Cancer Multi-Gene PCR Panel and the Oncomine Dx Target Test. However, routine pathological examination based primarily on hematoxylin and eosin (HE) staining has not typically led to the identification of driver gene mutations, with some exceptions, such as *ALK* fusion lung cancer [2,6]. We report a case of lung adenocarcinoma in which *MET* exon 14 skipping and *KRAS* mutations were suggested as potential driver gene mutations based on HE staining, thyroid transcription factor-1 (TTF-1; clone: 8G7G3/1) immunohistochemical staining, and other clinical pathological findings. An article by Hayashi et al. shows the relationship between thanatosomes, cytoplasmic clearing, or nuclear pleomorphism and *MET* exon 14 skipping or *KRAS* mutations [7]. During a routine health check, a 76-year-old nonsmoking Japanese woman was identified as having an abnormal shadow in her chest. CEA was elevated, at 7.6 ng/mL. A subsequent computed tomography scan revealed a nodular lesion in segment 3 of the right upper lobe (Figure 1, (**a**): axial section, (**b**): sagittal section) reveal a 2.3 cm irregularly contoured solid mass in the peripheral region of the right upper lobe (Segment 3). The tumor has poorly defined margins, with spiculation, notch signs noted, and pleural indentation. There is no evident calcification. A bronchoscopy conducted at our respiratory medicine department confirmed a diagnosis of adenocarcinoma. Further evaluation classified the cancer as cT1cN0M0, cStage IA3 (TNM, UICC 8th edition), leading to the performance of a right upper lobectomy.

**Figure 2 diagnostics-15-00894-f002:**
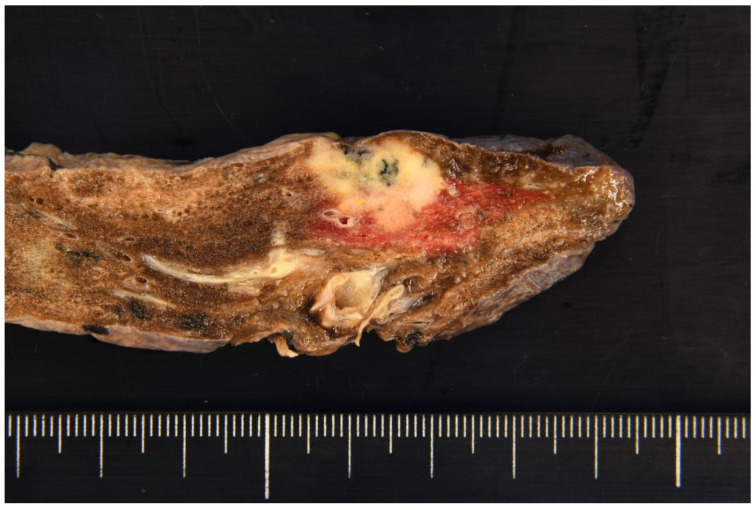
Macroscopically, the right upper lobectomy specimen weighing 110 g was submitted for pathologic examination. The tumor appeared lobulated, with a solid milky white appearance accompanied by areas of carbon deposition, and measured 19 mm × 15 mm × 10 mm. Notably, it exhibited features of pleural invasion.

**Figure 3 diagnostics-15-00894-f003:**
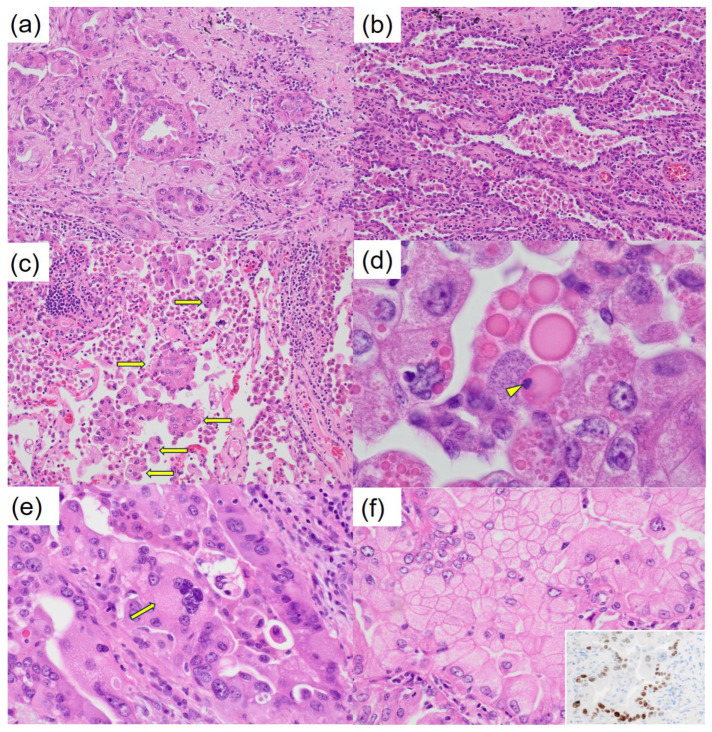
Microscopically, the tumor displayed a proliferation pattern consistent with adenocarcinoma, comprising 50% acinar pattern, 30% papillary pattern, and 20% lepidic pattern (**a**,**b**). Invasion of the pleura was observed, and the tumor was spread through the surrounding air spaces (**c**). An intrapulmonary micrometastasis was also identified near the primary lesion. Numerous hyaline globules (HGs), termed thanatosomes, were noted, with some containing nuclear dust (**d**). Nuclear pleomorphism was prominent (**e**), and some tumor cells showed scattered cytoplasmic clearing (**f**). Immunohistochemical staining for TTF-1 (clone: 8G7G3/1) reveals heterogeneous positive staining in the tumor cells (inset).

**Figure 4 diagnostics-15-00894-f004:**
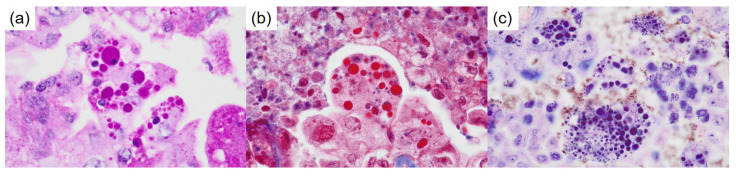
Special staining of the thanatosomes. (**a**) Periodic acid–Schiff reaction, (**b**) Masson’s trichrome stain, and (**c**) Mallory’s PTAH stain (×1000).

**Figure 5 diagnostics-15-00894-f005:**
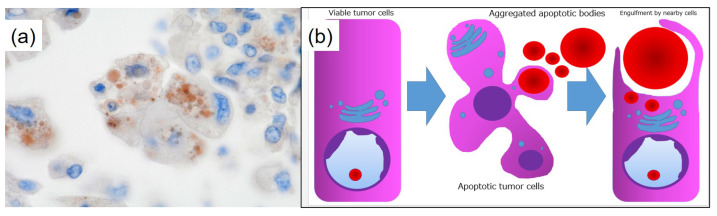
Staining and model of HG formation. (**a**) The thanatosomes (HGs) are positive for cleaved caspase-3 (×1000). Therefore, the globules are associated with apoptosis. There was no metastatic disease in the lymph nodes. The pathological staging was pT3N0M0; pStage IIB (TNM, UICC 8th edition). The patient’s postoperative course has been satisfactory. We ordered the Amoy Dx^®^ Lung Cancer Multi-Gene PCR Panel for further analysis. Genetic testing revealed a positive result for *KRAS* G12D/S and a negative result for *MET* exon 14 skipping. (**b**) A model for the formation of HGs in the present neoplasm. An intact tumor cell responds to a variety of injurious stimuli toward apoptosis. The apoptotic bodies are engulfed by neighboring viable cells. KRAS is a G protein associated with the RAS/MAPK signaling pathway. Gene alteration in the gene (chromosome 12p12.1) leads to a constitutively activated GTP-bound state, which is thought to drive the proliferation of cancer cells [8]. It is known that the frequency of driver gene mutations in lung adenocarcinoma differs between the USA and Japan. According to NCC Japan, the frequency of *KRAS* mutations is 9.7% among 319 Japanese cases of lung adenocarcinoma, making it the second most common driver gene mutation after *EGFR* [5]. *KRAS* mutations are observed in various types of adenocarcinomas, including HNF4α-positive mucinous adenocarcinoma, TTF1-positive terminal respiratory unit type adenocarcinoma, and epithelial-mesenchymal transition type non-terminal respiratory unit type adenocarcinoma [9,10]. The *KRAS* G12C mutation is associated with poor prognosis in pulmonary adenocarcinoma of stages I to III [11]. It is the most common *KRAS* mutation in 13–16% of pulmonary adenocarcinoma cases in Western populations [11,12]. In contrast, among Asian populations, it is observed in approximately 4% of lung cancer cases, with a higher frequency specifically in pulmonary adenocarcinoma, and it is correlated with smoking [13]. Recently, sotorasib (AMG 510) has been reported to induce tumor shrinkage in patients with lung cancer harboring *KRAS* G12C mutations [11,14]. In our case, detailed investigations concluded that the lung adenocarcinoma was due to a *KRAS* gene alteration. Thanatosomes, first proposed in 2000, were initially categorized as apoptosis-associated bodies (Figure 5b) [15] and can be associated with both tumor and non-tumor lesions. We reported instances in which not all thanatosomes were apoptosis-related bodies [16]. Thanatosomes reveal periodic acid–Schiff reactivity with diastase resistance, fuchsinophilia with Masson’s trichrome stain, and dark blue-black color with Mallory’s PTAH stain, as shown in Figure 4 [15,16]. Hayashi et al. reported that thanatosomes (HGs) were shown in 10.7%, nuclear pleomorphism in 13.1%, and clear cell features in 8.2% of *KRAS*-mutant pulmonary adenocarcinomas [7]. Thanatosomes, nuclear pleomorphism, and clear cell features were also observed in the present case (Figure 3). Hayashi et al. stated that thanatosomes associated with lung cancer are indeed apoptosis-related bodies [7]. Utilizing cleaved caspase-3 as an immunohistochemical marker for apoptosis, we also showed that thanatosomes are clearly apoptosis-associated bodies (Figure 5a) [16]. Cleaved caspase-3, an activated caspase-3, is an excellent and reproducible immunohistochemical marker for apoptosis. The biochemical pathways of apoptosis are controlled by caspases (cysteine aspartate-specific proteases), which cleave and activate various intracellular proteins. It functions as a control tower for apoptosis: it cleaves poly (ADP-ribose) polymerase, cytokeratin 18, vimentin, actin, and other intracytoplasmic proteins. It has been applied to detect apoptotic neoplastic cells in paraffin sections [15]. Histological findings in the present case revealed the presence of nuclear dust within some of the thanatosomes (Figure 3d), supporting this conclusion. To our knowledge, the present case report is the first to show that one notable morphological feature is the presence of thanatosomes in *KRAS*-mutant pulmonary adenocarcinoma. Building on the findings of Hayashi et al. [7], we propose that the three pathological findings of thanatosomes, nuclear pleomorphism, and cytoplasmic clearing are associated with *MET* exon 14 skipping and *KRAS* gene alterations in most but not all cases [7]. Pathologists observing these three histological features in lung adenocarcinoma specimens may consider conveying this information to clinicians, given the high prevalence of *KRAS* mutations in such cases. Thanatosomes, cytoplasmic clearing, and nuclear pleomorphism were unevenly distributed within the tumor. This is likely because tumors are not generally composed of a single clone but rather are made up of heterogeneous clones. Research on medical applications of generative artificial intelligence (AI) has advanced significantly in recent years. It is anticipated that standard implementations of AI in pathology are not far off. Multiple studies have shown that using convolutional neural networks makes it possible to predict the presence of various driver gene mutations solely from HE-stained images [17,18]. By continuing to compare HE-stained specimen images with driver gene mutations, as illustrated in this case report, we can incorporate these insights into AI systems. This could improve the accuracy in detecting driver gene mutations from HE-stained images, potentially shortening the turnaround time from specimen collection to the reporting of genetic test results to patients. This study has one major limitation: it is a single case report from a single institution in Japan. Additional clinicopathological analyses, including multicenter studies, are needed to determine the cause and pathophysiology of this condition. Therefore, reports from other countries, cultures, and hospitals are awaited. In conclusion, the authors conducted a detailed analysis of tissue diagnosis centered on HE-stained specimens, allowing them to estimate driver gene mutations in this study. Specifically, we identified three histological features as particularly important: HGs, nuclear pleomorphism, and cytoplasmic clearing. HGs are associated with apoptosis and suggest apoptotic activity in *KRAS*-mutant lung adenocarcinoma. Moving forward, we aim to utilize routine HE-stained tissue images to estimate various driver gene mutations as a contribution to improving the diagnosis and treatment of lung cancer.

## Data Availability

Not applicable.

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
