# Peer review of "Lung Adenocarcinoma Exhibiting Thanatosomes (Hyaline Bodies), Cytoplasmic Clearing, and Nuclear Pleomorphism, with a KRAS Mutation"

_diagnostics, 2025, doi:10.3390/diagnostics15070894_

Round 1

Reviewer 1 Report

Comments and Suggestions for Authors

This is an interesting case report of a KRAS-mutated lung adenocarcinoma with characteristic histology, including abundant hyaline globules. Although only one case is not sufficient to draw a conclusion in which these findings are specific to genetic alterations. There are several concerns regarding this. 

Major comments

The title should be descriptive because of the limitations of only one case. The characteristics of this case cannot be judged as a general finding of KRAS-mutated lung adenocarcinoma or not. The title should be more descriptive. Detailed histochemical and immunohistochemical studies of the hyaline bodies in this case were of interest.

Page 2, line 52:

In the main text, the reported case was suspected to have both MET exon14 skipping and KRAS mutations. Please add a description of the findings to suspect such driver genes and how (by AI?), who (pathologist).

The definition of “thanatosome” should be documented more in detail. Morphological characteristics of apoptosis include nuclear chromatin aggregation and fragmentation. The nuclei in Figure 5a do not show apoptotic features. This discrepancy is unacceptable because the authors insisted that hyaline bodies were related to apoptosis in this case.

Hyaline globules are often observed in lung cancers. In this case report, only one case was insufficient to conclude the relationship between KRAS mutation and morphology.

Minor comments

Line 197: The year 2026 may be a typographical error.

Comments on the Quality of English Language

Need native speaker editing

Author Response

Reviewer 1

Major comments

The title should be descriptive because of the limitations of only one case. The characteristics of this case cannot be judged as a general finding of KRAS-mutated lung adenocarcinoma or not. The title should be more descriptive. Detailed histochemical and immunohistochemical studies of the hyaline bodies in this case were of interest.

 →Thank you for your helpful comments. I have changed the title as follows: Lung Adenocarcinoma Exhibiting Thanatosomes (Hyalines Body), Cytoplasmic Clearing, and Nuclear Pleomorphism, with a KRAS Mutation: A Case Report

Page 2, line 52:

In the main text, the reported case was suspected to have both MET exon14 skipping and KRAS mutations. Please add a description of the findings to suspect such driver genes and how (by AI?), who (pathologist).

→Thank you for your helpful comments. We have added the following statement: “An article by Hayashi et al. shows the relationship between Thanatosomes, cytoplasmic clearing, or nuclear pleomorphism, and MET exon14 skipping or KRAS mutations [7].”

The definition of “thanatosome” should be documented more in detail. Morphological characteristics of apoptosis include nuclear chromatin aggregation and fragmentation. The nuclei in Figure 5a do not show apoptotic features. This discrepancy is unacceptable because the authors insisted that hyaline bodies were related to apoptosis in this case.

→ Reviewer 1 mentions that no clear images of apoptotic bodies (Thanatosomes) are shown in Figure 5a. However, the tissue findings are not uniform, and distinct areas, such as those seen in Figure 5d, were also observed. Tumors are inherently heterogeneous, and apoptotic activity is thought to vary depending on the location. Therefore, we have added the following sentences: “Thanatosomes, cytoplasmic clearing, and nuclear pleomorphism were unevenly distributed within the tumor. This is likely because tumors are not generally composed of a single clone but rather are made up of heterogeneous clones.”

Hyaline globules are often observed in lung cancers. In this case report, only one case was insufficient to conclude the relationship between KRAS mutation and morphology.

→Indeed, what Reviewer 1 is saying is valid. However, this is precisely why, in the limitations paragraph, we emphasize the need for further research across multiple centers in more countries.

Minor comments

Line 197: The year 2026 may be a typographical error.

→This case report is scheduled to be published as a paper in 2025, followed by a poster presentation at the 2026 Annual Meeting of the Japanese Society of Pathology. So, the information provided is not incorrect.

Comments on the Quality of English Language

Need native speaker editing

→The revised manus

Reviewer 2 Report

Comments and Suggestions for Authors
  1. The title of the article fully reflects the content of the article.
  2. In the “Abstract” section, the authors briefly presented the content of the article: the complexity of detecting mutations in lung adenocarcinoma was discussed and a report was presented, linking thanatosomes in KRAS-mutant pulmonary adenocarcinoma to apoptosis via cleaved caspase-3 staining. In addition, in the "Abstract" the authors mention artificial intelligence (AI) and indicate that convolutional neural networks can predict the presence of various driver gene mutations solely from hematoxylin and eosin stained images. Typically, the Abstract section presents information about your own research results and conclusions based on your own results. In my opinion, the reference to AI in the Abstract section should be removed.
  3. The "Keywords" presented in the article reflect the content of the article and are necessary. The keyword AI does not reflect the main results of the article, it is used in the discussion. AI should be removed.
  4. The main content of the article follows. It is noteworthy that the relevance of the study, description of the clinical case, discussion and conclusions are presented in a single text. The description of the patient was accompanied by 5 figures. It is noteworthy that the purpose of the study is not presented clearly.
  5. All the results of the study are important and necessary. The results of the study are discussed using the literature, and the authors pointed out the prospects for using artificial intelligence in processing clinical research data. It is important that the authors voiced the limitations of their study. I agree that additional clinical and pathological analyses, including multicenter studies, are needed to finally determine the cause and pathophysiology of this condition. Conclusions are presented at the end of the text.
  6. The article is important for clinical medicine, specifically for the diagnosis of patients with lung adenocarcinoma. The text of the article is written clearly. The manuscript did not raise any ethical issues. All references to publications in the References section are necessary and correct, and written in the correct style. I have no concerns about the similarity of this article with other articles published by the same authors.
  7. I strongly recommend dividing the main text of the article into sections “Introduction”, “Detailed description of the case”, “Discussion”, and “Conclusions”.
  8. Competing interests of the authors do not create bias in the presentation of results and conclusions.

Author Response

Reviewer 2

Comments and Suggestions for Authors

  1. The title of the article fully reflects the content of the article.

→Thank you for your kind feedback. Based on a comment made by Reviewer 1, I have changed the title as follows: “Lung Adenocarcinoma Exhibiting Thanatosomes (Hyaline Bodies), Cytoplasmic Clearing, and Nuclear Pleomorphism, with a KRAS Mutation: A Case Report.”

  1. In the “Abstract” section, the authors briefly presented the content of the article: the complexity of detecting mutations in lung adenocarcinoma was discussed and a report was presented, linking thanatosomes in KRAS-mutant pulmonary adenocarcinoma to apoptosis via cleaved caspase-3 staining. In addition, in the "Abstract" the authors mention artificial intelligence (AI) and indicate that convolutional neural networks can predict the presence of various driver gene mutations solely from hematoxylin and eosin stained images. Typically, the Abstract section presents information about your own research results and conclusions based on your own results. In my opinion, the reference to AI in the Abstract section should be removed.

→Thank you for your kind feedback. I removed all reference to AI from the abstract.

  1. The "Keywords" presented in the article reflect the content of the article and are necessary. The keyword AI does not reflect the main results of the article, it is used in the discussion. AI should be removed.

→Thank you for your feedback. I removed “AI” from the keywords.

  1. The main content of the article follows. It is noteworthy that the relevance of the study, description of the clinical case, discussion and conclusions are presented in a single text. The description of the patient was accompanied by 5 figures. It is noteworthy that the purpose of the study is not presented clearly.

→Thank you for your feedback. As this paper is a case report, I believe it is unnecessary to include a clear research objective, as would be required in experimental studies.

  1. All the results of the study are important and necessary. The results of the study are discussed using the literature, and the authors pointed out the prospects for using artificial intelligence in processing clinical research data. It is important that the authors voiced the limitations of their study. I agree that additional clinical and pathological analyses, including multicenter studies, are needed to finally determine the cause and pathophysiology of this condition. Conclusions are presented at the end of the text.

→Thank you for agreeing with the authors’ arguments.

  1. The article is important for clinical medicine, specifically for the diagnosis of patients with lung adenocarcinoma. The text of the article is written clearly. The manuscript did not raise any ethical issues. All references to publications in the References section are necessary and correct, and written in the correct style. I have no concerns about the similarity of this article with other articles published by the same authors.

→Thank you very much.

  1. I strongly recommend dividing the main text of the article into sections “Introduction”, “Detailed description of the case”, “Discussion”, and “Conclusions”.

→Thank you for your feedback. In the “Interesting Images” section of Diagnostics, it is customary not to divide the content into sections such as “Introduction,”” Case Details,”” Discussion,” and “Conclusion.”

  1. Competing interests of the authors do not create bias in the presentation of results and conclusions.

→Thank you very much.